# Sustained Increase in Serum Glial Fibrillary Acidic Protein after First ST-Elevation Myocardial Infarction

**DOI:** 10.3390/ijms231810304

**Published:** 2022-09-07

**Authors:** Jan Traub, Katja Grondey, Tobias Gassenmaier, Dominik Schmitt, Georg Fette, Stefan Frantz, Valérie Boivin-Jahns, Roland Jahns, Stefan Störk, Guido Stoll, Theresa Reiter, Ulrich Hofmann, Martin S. Weber, Anna Frey

**Affiliations:** 1Medical Clinic and Policlinic 1, University Hospital Würzburg, 97080 Würzburg, Germany; 2German Comprehensive Heart Failure Center, University and University Hospital Würzburg, 97080 Würzburg, Germany; 3Institute of Neuropathology, University Medical Center Göttingen, 37075 Göttingen, Germany; 4Department of Diagnostic and Interventional Radiology, University Hospital Würzburg, 97080 Würzburg, Germany; 5Data Integration Center, University Hospital Würzburg, 97080 Würzburg, Germany; 6Institute for Virology and Immunobiology, University of Würzburg, 97080 Würzburg, Germany; 7Interdisciplinary Bank of Biomaterials and Data Würzburg, University and University Hospital Würzburg, 97080 Würzburg, Germany; 8Department of Neurology, University Hospital Würzburg, 97080 Würzburg, Germany; 9Department of Neurology, University Medical Center Göttingen, 37075 Göttingen, Germany; 10Fraunhofer-Institute for Translational Medicine and Pharmacology, 37075 Göttingen, Germany

**Keywords:** myocardial infarction, STEMI, glial fibrillary acidic protein, GFAP, neurofilament light chain, NfL, glial damage, cardiac magnetic resonance imaging, MRI, infarction size

## Abstract

Acute ischemic cardiac injury predisposes one to cognitive impairment, dementia, and depression. Pathophysiologically, recent positron emission tomography data suggest astroglial activation after experimental myocardial infarction (MI). We analyzed peripheral surrogate markers of glial (and neuronal) damage serially within 12 months after the first ST-elevation MI (STEMI). Serum levels of glial fibrillary acidic protein (GFAP) and neurofilament light chain (NfL) were quantified using ultra-sensitive molecular immunoassays. Sufficient biomaterial was available from 45 STEMI patients (aged 28 to 78 years, median 56 years, 11% female). The median (quartiles) of GFAP was 63.8 (47.0, 89.9) pg/mL and of NfL 10.6 (7.2, 14.8) pg/mL at study entry 0–4 days after STEMI. GFAP after STEMI increased in the first 3 months, with a median change of +7.8 (0.4, 19.4) pg/mL (*p* = 0.007). It remained elevated without further relevant increases after 6 months (+11.7 (0.6, 23.5) pg/mL; *p* = 0.015), and 12 months (+10.3 (1.5, 22.7) pg/mL; *p* = 0.010) compared to the baseline. Larger relative infarction size was associated with a higher increase in GFAP (ρ = 0.41; *p* = 0.009). In contrast, NfL remained unaltered in the course of one year. Our findings support the idea of central nervous system involvement after MI, with GFAP as a potential peripheral biomarker of chronic glial damage as one pathophysiologic pathway.

## 1. Introduction

Acute myocardial infarction (MI) remains a challenging condition and affects about 16 million patients worldwide every year [1]. It predisposes one not only to cardiac diseases, such as chronic heart failure, but also significantly increases the risk for cognitive impairment, including dementia and Alzheimer’s disease [2,3]. Furthermore, patients experiencing an acute MI have a three times higher risk of developing depression [4], and about one third suffer from a major depressive episode one year post-MI [5]. Both cognitive impairment and depression are of great relevance for the individual patient, as they increase mortality and the risk of new cardiovascular events [6]. Mechanistically, microvascular dysfunction, oxidative stress, and neuro-humoral activation seem to play an essential role in this regard [7]. Recently, it has been postulated that neuroinflammation after MI might also explain central nervous system involvement [8], and that ^11^C-methionine identifies an astrocyte-driven component of neuroinflammation after acute MI [9]. Following this line, we wondered if there is an alteration of lately emerging peripheral serum biomarkers of glial and neuronal damage after MI. Serum glial fibrillary acidic protein (GFAP), as an intermediate filament of mature astrocytes, associates with cognitive decline in neurological disorders such as Alzheimer’s disease, reflecting disease-related reactive astrogliosis [10] and astroglial damage in the brain [11,12]. Our group has recently shown that memory deficits in patients with chronic heart failure associate with elevated levels of serum GFAP [13]. Furthermore, neurofilament light chain (NfL) as a marker for neuronal damage has gained attention in predicting pre-clinical and clinical dementia [14,15,16]. Serum levels of NfL correlate with brain volume loss and cognitive deficits in the general population [17]. In the present study, of 45 well-characterized patients with ST-elevation MI, we, for the first time, describe the course and predictors of both biomarkers for glial and neuronal damage GFAP and NfL within the 12 months after the index event.

## 2. Results

### 2.1. Characteristics of Included Patients

Within the locally included patients, sufficient biomaterial (serum) at study inclusion, which was 0–4 days after the index event, and at three follow-up visits after 2, 6, and 12 months was available from 45 patients after their first ST-elevation MI (STEMI), as confirmed by coronary angiography. Their age was 56 ± 11 years (ranging from 28 to 78 years), and 11% were female (Table 1). Apart from two, all patients (95.6%) underwent stent implantation. The most frequent culprit vessel was the left anterior descending artery (42.2%), followed by the right coronary artery (35.6%), and the circumflex artery (22.2%). The median (quartiles) peak creatine kinase was 560 (302, 1136) U/l and high-sensitive Troponin T peaked at 2044 (950, 2952) pg/nl. Importantly, no patient died during the one-year follow-up. Cardiac magnetic resonance imaging (cMRI) evaluation revealed a mean left ventricular ejection fraction of 52.1 ± 8.1% early after the index event (day 7–9 after STEMI). Median relative infarction size significantly decreased from 15.5 (9.4, 20.0)% to 8.5 (6.2, 14.8)% one year after MI (*p* < 0.001).

### 2.2. Course of GFAP and NfL after First STEMI

We collected baseline serum samples at study inclusion, which was 0–4 days after the index event. At this timepoint, we measured GFAP serum levels of 63.8 (47.0, 89.9) pg/mL, ranging from 17 to 217 pg/mL. Within 3 months after MI, GFAP significantly (*p* = 0.007) increased with a median change of +7.7 (0.4, 19.4) pg/mL (Figure 1). Thereafter, no further changes were found, but GFAP remained elevated compared to baseline after 6 months (+11.7 (0.6, 23.5) pg/mL; *p* = 0.015) and 12 months (+10.3 (1.5, 22.7) pg/mL; *p* = 0.010). Consequently, 36 patients (80%) had elevated GFAP levels one year after STEMI compared to early after MI.

The median serum NfL at baseline was 10.6 (7.2, 14.8) pg/mL and ranged from 4.4 to 55.2 pg/mL. Longitudinal analysis of individual patients revealed a non-significant median change of −0.6 (−2.6, 1.5) pg/mL after 3 months, −0.5 (−2.0, 1.2) pg/mL after 6 months, and −0.4 (−2.5, 1.0) pg/mL 12 months, compared to baseline levels. Thus, we could not detect relevant changes in serum NfL in the year after STEMI.

Notably, GFAP and NfL levels interrelated with one another at baseline (ρ = 0.47; *p* < 0.001) and 12 months after MI (ρ = 0.57; *p* < 0.001). ΔGFAP related to ΔNfL (ρ = 0.32; *p* = 0.030) one year after the index event.

### 2.3. Clinical Correlates

When we compared patients with a GFAP increase to those without (Table 1), relative infarction size was the only parameter, which was significantly different between groups. Along with a larger infarction size, patients with a GFAP increase also tended to have higher peak values of troponin T, creatine kinase, and lactate dehydrogenase, but these trends were not significant.

To further identify clinical correlates of both absolute levels and changes in GFAP and NfL, we performed Spearman’s correlation analysis with clinical, laboratory, and radiological parameters early after STEMI (Table 2). Patient age strongly correlated to both GFAP and NfL levels at the baseline and after 3 months (ρ = 0.73/0.56), 6 months (ρ = 0.76/0.57), and 12 months (ρ = 0.74/0.61). The absolute values of both biomarkers did not correlate to any of the other examined parameters.

Regarding changes in biomarker levels, ΔGFAP correlated to the relative infarction size determined in cMRI early after STEMI (Figure 2). Peak creatine kinase levels emerged as the only significant predictor of ΔNfL in the course of 12 months after STEMI. To exclude that changes in other laboratory parameters are involved in the reported increase in GFAP, we analyzed how available values changed in the year after MI (Table 3): Among others, we detected increases in hemoglobin and albumin and decreases in C-reactive protein and glomerular filtration rate. Importantly, these changes did not correlate to changes in GFAP (ΔGFAP) one year after MI.

## 3. Discussion

This post hoc measurement of recently established glial (GFAP) and neuronal (NfL) serum biomarkers after acute MI revealed the following main findings: first, GFAP as a glial biomarker representing glial damage, but not NfL representing neuronal damage, increased in the first 3 months after myocardial ischemia and remained elevated within the year after MI; second, relative infarction size significantly correlated to GFAP increase.

This evaluation was based on a total of 45 STEMI patients between 28 and 78 years. All of them survived the one-year follow-up. In general, one-year survival probability ranges between 98% and 64% after STEMI and is strongly dependent on the patient’s age [18,19]. This discrepancy is most probably due to the highly preselected patient cohort in this study; it included patients who were clinically stable early after MI and were able to undergo a prolonged 2 h MRI protocol. Patients with cardiogenic shock, any rhythmological status, and resuscitation were excluded. Thus, hypoxic brain damage and the concomitant alteration of biomarker levels due to hypoperfusion are highly unlikely.

For GFAP, we determined the median serum levels of 64 (47, 90 pg/mL) at study inclusion 0–72 h after STEMI, which are comparable to published cohorts. Others reported a median GFAP of 69 pg/mL and 92 pg/mL in healthy individuals and found a strong age-dependency regardless of disease state, as we did in our study [20,21]. The median serum concentration of NfL early after the index event was 10.6 (7.2, 14.8) pg/mL. Comparable studies in normal aging find median NfL concentrations of 22 and 23 pg/mL in the age range 50–60 years, regarding the mean age of 56 years in our cohort [17,20]. However, the strong correlation to age at all four time points fully matches with previous reports in healthy and neurologically diseased patients [22,23,24]. It is thought that this increase is associated with neuronal loss in the normal aging population and does not represent a pathology per se [22].

To our knowledge, this is the first report about the course of neuronal and glial serum biomarkers after STEMI, without reanimation or cardiac arrest. There are numerous reports about NfL as an early and sensitive predictor of long-term neurological outcomes in patients after cardiac arrest [25,26,27]. NfL concentrations <200 pg/mL, as they were in our cohort, predicted a good neurological outcome [25]. Similar findings for GFAP foster the idea of relevant glial and neuronal damage in acute hypoxia-induced brain injury [28,29]. Interestingly, GFAP levels peaked at 24 h after cardiac arrest before decreasing from day to day after brain hypoxia, while long-term data are missing [30]. This implies that the differential and sustained increase in GFAP within the first 3 months of our study might not have been due to an acute hypoxic event, but rather caused by chronic processes within the central nervous system after MI. Recently published serum GFAP elevations in chronic heart failure patients and independent relations to memory impairment [13] further support the concept that the increased risk for AD after MI [2] might be triggered by chronic glial activation or damage.

Advances in whole-body imaging after MI using positron emission tomography suggest acute and chronic neuroinflammation after MI [8,9]. Here, mitochondrial translocator protein served as a marker for microglial activation [8]. It is known that the activation of microglia into their inflammatory M1 state triggers the subsequent activation of neighboring astrocytes [31]. Recently, the same group reported that ^11^C-methionine, a marker of inflammation, co-localized with astrocytes in mice after MI [9]. Furthermore, they found an elevated density of GFAP-positive astrocytes in the hippocampus three days after experimental MI compared to integrin-treated animals [9]. Thus, the observed increase in serum GFAP might be a result of astroglial activation in the context of neuroinflammation after MI. Gliosis in the amygdala of rats following MI further solidifies this idea [32]. MI also coincides with pro-inflammatory alterations in the brain microvasculature, as shown in autopsied patients [33], and tumor necrosis factor alpha mediated disruption of the blood–brain barrier [4]. In the context of the neurovascular unit, this endothelial dysfunction might be caused by astroglial activation after MI or trigger (micro-) glial activation itself [31]. Interestingly, a recent study found that plasma S100 calcium-binding protein B (S100B) levels increased significantly in acute MI patients compared to the levels of stable angina pectoris patients and control subjects [34]. As S100B is also considered a peripheral marker of blood–brain barrier disruption [35], these findings may suggest that a disruption of this structure after MI causes the reported increase in GFAP.

There are reports that suggest that inflammation after MI occurs in brain nuclei that play key roles in cardiovascular regulation [36] and MI induces hypothalamic cytokine synthesis [37]. In this line, the recently postulated activation of astrocytes in the hypothalamus paraventricular nucleus [38] may explain the increase in serum GFAP in our study. Finally, it is thought that inflammasome-mediated neurodegeneration after MI [39] predisposes one to neurodegenerative disorders [2] and changes in the spatial distribution of the Purkinje network [40]. There are even cases of cortical laminar necrosis after MI [41].

To establish a potential connection between MI and GFAP increase, we analyzed cardiac correlates of ΔGFAP. Indeed, a higher relative infarction size early after MI was correlated with an increase in GFAP (Figure 2). One might speculate that a higher inflammatory burden in larger infarctions might trigger more pronounced neuroinflammation, which goes along with a higher GFAP increase. In fact, infarction size was significantly correlated with peak C-reactive protein (ρ = 0.49; *p* = 0.002) in our cohort. Further mechanisms might include the more severe impairment of cerebral blood flow [42] or higher neuro-humoral activation in greater infarctions [43,44]. We also demonstrate that patients with an increase in the neuronal protein NfL (*n* = 19; 42% of all patients) have higher peak values of creatine kinase, potentially suggesting secondary axonal damage in large infarctions.

Taken together, we here propose a differential increase in the glial biomarker GFAP in the first year after MI, which might relate to infarction size. A particular strength of our study is the detailed cMRI analysis of every patient, which allowed an accurate determination of infarction size and ejection fraction. The limitations of this investigation include the relatively small patient number, which was not large enough to fully account for potentially confounding factors, such as co-morbidities, the success of interventions, and post-intervention complications. Furthermore, blood samples early after MI (baseline) were not taken on exactly the same day for all patients. Regarding the short serum half-life of GFAP and NfL [45], future serial sampling might unveil dynamics early after MI more precisely.

The most important limitation is the absence of neurological and neuropsychological testing and brain imaging in this cohort. These would provide further insight into the clinical relevance of the detected GFAP increase. Therefore, we think that future studies addressing brain morphology (using cerebral MRI or positron emission tomography) and cognitive function are urgently needed to further decipher the here suggested glial damage after MI and evaluate the role of peripheral biomarkers in this regard.

## 4. Materials and Methods

### 4.1. Study Design

All included patients were participants in the investigator-initiated, prospective, multicenter diagnostic ETiCS study (Etiology, Titre-Course, and effect on Survival) carried out in Würzburg within the German Competence Network Heart Failure. The ETiCS study was conducted according to the Declaration of Helsinki and followed Good Clinical Practice standards under the supervision of an independent Data Monitoring and Endpoint Committee. The Ethics Committee of the University of Würzburg approved the study (Vote 186/07), and all patients gave written, informed consent.

### 4.2. Selection of Patients

Patients with a first ST-elevation myocardial infarction, defined according to AHA/ACC Guidelines from 2004 [46], who were experiencing typical symptoms and were above the age of 18 years, entered into the study between May 2010 and May 2017 as reported earlier [47]. Exclusion criteria comprised comorbidities that influence the immune system or require immunosuppressive treatment (Table 4). For our analysis, we selected patients with full sets of cardiac MRIs over the follow-up of one year and follow-up blood samples, including the storage of biomaterial including serum.

### 4.3. Study Flow

Patients entered into the study within the first 72 h after hospital admission (median 1.2 days after admission). The data collected at the baseline included medical history, physical examination, current medication, and records from cardiac catheterization and coronary angiography. All patients had cardiac magnetic resonance imaging (cMRI) between day 7 and 9 after STEMI and a follow-up cMRI after 12 months (±14 days). Blood samples were taken according to the study protocol at inclusion and repeatedly during the in-patient stay. They were collected for routine clinical chemistry investigations at the certified facility of the University Hospital Würzburg.

### 4.4. Ultrasensitive Immunoassays for GFAP/NfL

Non-fasting participants were positioned seated for at least 5 min before the puncture of venous blood samples. Serum samples were processed immediately, i.e., remained at room temperature for 30 min and were centrifuged for 10 min at 2000× *g*. The serum was stored at −80 °C until analysis. Serum GFAP and NfL were measured using the Simoa Human Neurology 2-Plex kit (103520; Quanterix^TM^, Billerica, MA, USA) on a Simoa HD-1 Analyzer instrument (Quanterix^TM^, Billerica, MA, USA) in accordance with the manufacturer’s instructions. These measurements were performed blind to the patients’ other results at the Institute of Neuropathology at the University Medical Center Göttingen.

### 4.5. Cardiac Magnetic Resonance Imaging

A 3.0 Tesla MRI scanner (MAGNETOM Trio and MAGNETOM Prisma, Siemens Healthineers, Erlangen, Germany) was used for both the baseline and the one-year follow-up cMRI. The protocol included a full stack of short-axis cine images from the mitral valve plane to the apex and three steady-state free precession cine images in the long axis. A single observer with >5 years of experience in cMRI analyzed the images using commercially available certified cMRI evaluation software (cmr42, Circle Cardiovascular Imaging, Calgary, AB, Canada). He was blinded to all further data. Endocardial and epicardial contours were manually delineated on all short-axis slices. Left ventricular end-diastolic, end-systolic, stroke volume, left ventricular ejection fraction, and left ventricular mass were calculated from the short-axis cine images. The volume was multiplied by the standard for myocardial density (1.05 g/mL) to obtain myocardial mass. Infarct size was determined using late gadolinium enhancement on short-axis images. To identify infarcted myocardium (defined as signal intensity >5 standard deviations over a region of interest placed in the healthy myocardium), a semi-automated threshold detection was used. Hypo-intense areas, representing microvascular obstruction, were manually included. Relative infarction size was defined as the percentage of infarction mass by systolic left ventricular mass. More details regarding cMRI have been published previously [47].

### 4.6. Statistical Evaluation

Data are expressed as medians (quartiles) or *n* (percentage), as appropriate. Group-wise comparisons were performed using an ANOVA or Kruskal–Wallis test, as appropriate. For paired samples, we used a Wilcoxon matched-pairs signed-ranks test due to non-normally distributed data. Peak blood values for analysis were selected from the values within the first nine days after MI. All data analyses were performed on a complete case basis. Missing values were not imputed; thus, the number of observations in each analysis depended on the availability of data for all variables. We identified correlates of the biomarker serum concentrations using Spearman’s correlation. All tests were two-tailed and should be considered exploratory. Hence, no correction for multiple testing was introduced. A probability of *p* < 0.05 was considered statistically significant. Statistical analysis and preparation of illustrations were performed using IBM SPSS Statistics 28.

## Figures and Tables

**Figure 1 ijms-23-10304-f001:**
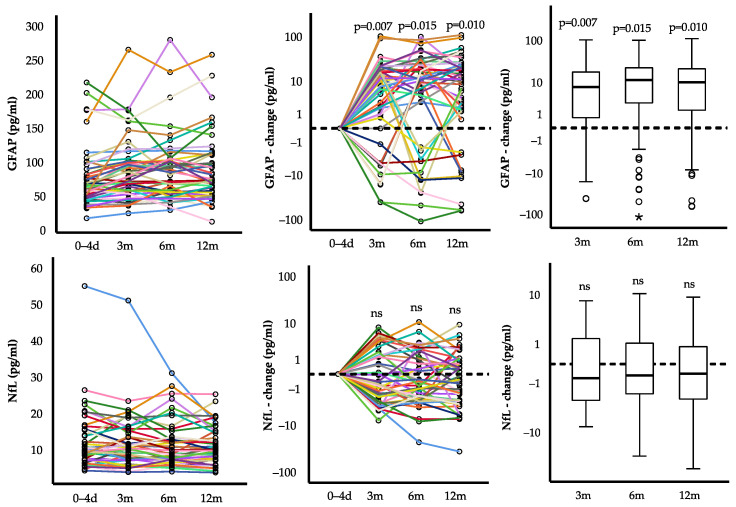
Course of serum GFAP (**top**) and NfL (**bottom**) serum levels after first myocardial infarction. Different colored lines connect the values of individual patients. Each dot represents the value of a patient at one time point. Wilcoxon signed-rank test (*n* = 45). * = outlier.

**Figure 2 ijms-23-10304-f002:**
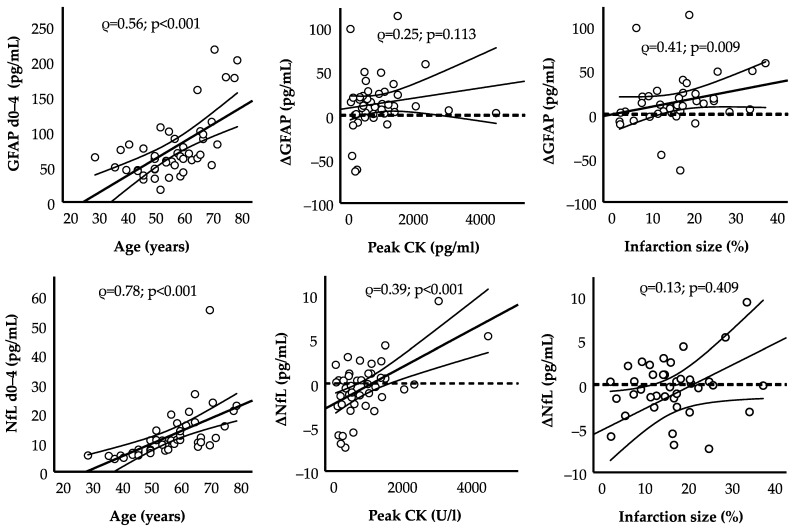
Spearman’s correlation of serum biomarkers and change over 12 months (Δ). Dots represent single patients (*n* = 45). CK = creatine kinase; ρ = Spearman’s rho. Solid trend lines and standard deviation (thin lines) are shown.

**Table 1 ijms-23-10304-t001:** Patient characteristics according to GFAP increase or decrease 12 months after STEMI.

	All Patients (*n* = 45)	ΔGFAP ≤ 0 (*n* = 9)	ΔGFAP > 0 (*n* = 36)	*p*
Age (years)	56 (49, 65)	56 (51, 66)	55 (46, 65)	0.399
Female sex (%)	5 (11.1%)	1 (11.1%)	4 (11.1%)	>0.999
Body mass index (kg/m^2^)	27 (24, 29)	26 (24, 30)	27 (24, 29)	0.744
Arterial hypertension *	17 (37.8%)	4 (44.4%)	13 (36.1%)	0.645
Diabetes mellitus ^†^	7 (15.6%)	2 (22.2%)	5 (13.9%)	0.537
Dyslipidemia ^‡^	8 (17.8%)	1 (11.1%)	7 (19.4%)	0.559
Current smoker	17 (37.8%)	1 (11.1%)	16 (44.4%)	0.181
Previous coronary artery disease	10 (22.2%)	2 (22.2%)	8 (22.2%)	>0.999
Previous myocardial infarction	0 (0.0%)	0 (0.0%)	0 (0.0%)	>0.999
Atrial fibrillation	2 (4.4%)	1 (11.1%)	1 (2.8%)	0.497
Peripheral vascular disease	4 (8.9%)	0 (0.0%)	4 (11.1%)	0.295
Pain-to-balloon time (min)	280 (123, 592)	210 (114, 784)	285 (131, 593)	0.532
Culprit vessel:				
- Left anterior descending	19 (42.2%)	4 (44.4%)	15 (41.7%)	0.880
- Circumflex artery	10 (22.2%)	2 (22.2%)	8 (22.2%)	>0.999
- Right coronary artery	16 (35.6%)	3 (33.3%)	13 (36.1%)	0.876
Stent implantation	43 (95.6%)	9 (100.0%)	34 (94.4%)	0.469
Classification of coronary artery disease				
- 1-vessel	27 (60.0%)	4 (44.4%)	23 (63.9%)	0.287
- 2-vessel	14 (31.1%)	3 (33.3%)	11 (30.6%)	0.872
- 3-vessel	4 (8.9%)	2 (22.2%)	2 (5.6%)	0.116
Peak creatine kinase (U/L)	560 (302, 1136)	239 (122, 659)	717 (402, 1326)	0.096
Peak troponin T (pg/nL)	2045 (950, 2952)	1334 (452, 2447)	2395 (1000, 3372)	0.145
Peak lactate dehydrogenase (U/L)	530 (366, 657)	372 (236, 607)	533 (375, 734)	0.117
End-diastolic volume (mL/m^2^) d 7–9	87 (80, 97)	90 (83, 94)	86 (79, 100)	0.924
Left ventricular ejection fraction (%) d 7–9	54 (46, 58)	55 (43, 59)	54 (47, 58)	0.836
Relative infarction size (%) d 7–9	15.5 (9.4, 20.0)	10.4 (2.7, 16.1)	16.2 (11.1, 21.6)	0.047
End-diastolic volume (mL/m^2^) 12 m	84 (74, 96)	85 (75, 104)	84 (73, 96)	0.593
Left ventricular ejection fraction (%) 12 m	53 (47, 59)	57 (44, 62)	53 (47, 58)	0.716
Relative infarction size (%) 12 m	8.5 (6.2, 14.8)	7.2 (2.4, 15.0)	8.5 (6.6, 14.8)	0.387

* = Sitting blood pressure > 140/90 mmHg or history of hypertension; ^†^ = History of Diabetes mellitus type II or HbA1c > 6.5; ^‡^ = Hyperlipidemia or statin treatment.

**Table 2 ijms-23-10304-t002:** Spearman’s correlation of clinical, laboratory, and cMRI parameters with NfL and GFAP serum levels. * = *p* < 0.05.

Spearman’s Rho (ρ)	GFAP 0–4 d (pg/mL)	GFAP 12 m (pg/mL)	ΔGFAP (pg/mL)	NfL 0–4 d (pg/mL)	NfL 12 m (pg/mL)	ΔNfL (pg/mL)
Age (years)	0.56 *	0.61 *	0.15	0.78 *	0.74 *	−0.15
Systolic blood pressure at admission (mmHg)	−0.19	0.04	0.23	−0.14	0.02	0.18
Diastolic blood pressure at admission (mmHg)	−0.14	0.01	0.11	−0.20	−0.08	0.18
Heart rate at admission (1/min)	−0.14	−0.13	0.10	−0.15	−0.08	0.11
Pain-to-balloon time (min)	0.13	0.10	−0.06	−0.16	−0.18	−0.09
Peak creatine kinase (U/L)	−0.14	0.03	0.25	−0.19	0.04	0.39 *
Peak troponin T (pg/mL)	0.08	0.10	0.03	0.00	0.08	0.00
Peak lactate dehydrogenase (U/L)	0.05	0.14	0.17	−0.03	0.07	0.08
Peak C-reactive protein (mg/dL)	0.19	0.13	−0.11	0.23	0.09	−0.23
Peak leucocytes (10^3^/µL)	−0.17	−0.10	0.11	−0.04	0.02	0.00
Normalized end-diastolic volume (mL/m^2^) d 7–9	0.08	0.02	−0.07	−0.09	−0.15	−0.14
Left-ventricular ejection fraction (%) d 7–9	−0.24	−0.21	0.02	−0.18	−0.26	−0.06
Relative infarction size (%) d 7–9	−0.02	0.19	0.41 *	0.05	0.16	0.13

**Table 3 ijms-23-10304-t003:** Course of laboratory parameters after first STEMI and correlation to ΔGFAP. * = significant change in Wilcoxon matched-pairs signed-ranks test.

	Day 0–4	12 Months	Δ12 m—d 0–4	~ΔGFAP	*p* Value
Glial fibrillary acidic protein (pg/mL)	64 (47, 90)	73 (53, 113)	10 (1, 23) *	-	
Neurofilament light chain (pg/mL)	11 (7, 15)	10 (7, 14)	0 (−2, 1)	0.32 *	0.030
Natrium (mmol/L)	139 (137, 140)	140 (138, 141)	1 (−1, 2) *	−0.14	ns
Potassium (mmol/L)	4.2 (4.0, 4.4)	4.4 (4.2, 4.6)	0.2 (0.0, 0.4) *	0.03	ns
Estimated GFR (mL/min/1.73 m^2^)	91 (80, 106)	82 (72, 99)	−7 (−14, 3) *	−0.14	ns
Urea (mg/dL)	31 (25, 38)	31 (26, 37)	0 (−7, 7)	0.13	ns
Uric acid (mg/dL)	5.2 (4.5, 6.0)	6.0 (5.0, 6.6)	0.7 (0.0, 1.4) *	0.17	ns
Cholesterol (mg/dL)	190 (165, 211)	161 (145, 190)	−25 (−60, 11) *	−0.08	ns
Low-density lipoprotein (mg/dL)	95 (78, 114)	82 (66, 101)	−11 (−42, 5) *	0.03	ns
High-density lipoprotein (mg/dL)	39 (33, 46)	45 (38, 59)	5 (2, 9) *	−0.02	ns
Triglycerides (mg/dL)	128 (105, 162)	162 (98, 195)	45 (−12, 108) *	−0.14	ns
Aspartate aminotransferase (U/L)	48 (33, 93)	28 (21, 32)	−21 (−155, −4)	−0.07	ns
Alanine aminotransferase (U/L)	45 (31, 65)	30 (20, 34)	−19 (−27, 3)	−0.23	ns
γ-glutamyltransferase (U/L)	44 (28, 70)	32 (19, 38)	−3 (−53, 4)	−0.22	ns
Lactate dehydrogenase (U/L)	530 (366, 657)	198 (175, 240)	−293 (−491, −108) *	−0.15	ns
Creatine kinase (U/L)	560 (302, 1136)	113 (82, 165)	−426 (−975, −95) *	−0.19	ns
NT-proBNP (pg/mL)	844 (627, 1368)	133 (48, 291)	−755 (−1329, −403) *	−0.02	ns
Troponin T (pg/mL)	2044 (950, 2952)	6 (5, 9)	−2037 (−2946, −945) *	−0.03	ns
Complement factor C3c (mg/dL)	121 (107, 132)	118 (105, 131)	−3 (−14, 8)	−0.02	ns
Albumin (g/dL)	4.2 (3.9, 4.3)	4.6 (4.4, 4.9)	0.5 (0.3, 0.8) *	0.00	ns
Hemoglobin (g/dL)	13.9 (13.0, 15.1)	14.6 (13.8, 15.5)	0.8 (0.2, 1.7) *	−0.09	ns
Hematocrit (%)	41 (38, 43)	43 (41, 45)	3 (0, 5) *	−0.16	ns
Thrombocytes (10^3^/µL)	242 (189, 288)	235 (189, 285)	3 (−23, 19)	0.12	ns
Leucocytes (10^3^/µL)	9.2 (8.1, 10.9)	6.3 (5.2, 7.2)	−2.8 (−4.2, −1.9) *	−0.14	ns
HbA1c (%)	5.6 (5.4, 6.7)	5.7 (5.4, 6.0)	0.0 (−0.2, 0.3)	0.15	ns
C-reactive protein (mg/dL)	1.7 (0.7, 6.2)	0.1 (0.0, 0.2)	−1.6 (−6.1, −0.6) *	0.12	ns

**Table 4 ijms-23-10304-t004:** Inclusion and exclusion criteria of the ETiCS study.

Inclusion Criteria
Age ≥ 18 years and written informed consent
No previous history of myocardial infarction
New onset of chest pain in the past 7 days
Confirmed myocardial infarction in cardiac catheterization
Fulfillment of STEMI criteria:
- Either ST-segment elevations in at least two adjacent leads
- ST-segment elevations ≥ 0.1 mV in the extremity leads
- ST-segment elevations ≥ 0.2 mV in the chest wall leads
- Or new onset left bundle branch block with matching clinic
Exclusion criteria
Tumor disease or other critical illness with a life expectancy of <1 year
Terminal renal failure or hemodialysis
Rheumatologic disease with the need for immunomodulatory drugs
Autoimmune disease with the need for immunomodulatory drugs
Congenital neuromuscular disease
Myasthenia gravis
Graves’ disease
Incapacity to consent
Sociological, psychological, mental, or other limitations
Continued alcohol or drug abuse
Pregnancy or lactation

## Data Availability

The data presented in this study are available on request from the corresponding author.

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
