# Peer review of "Sustained Increase in Serum Glial Fibrillary Acidic Protein after First ST-Elevation Myocardial Infarction"

_ijms, 2022, doi:10.3390/ijms231810304_

Round 1

Reviewer 1 Report

The current manuscript describes one year follow up of patients with first ST-elevation myocardial infarction. The degree of change in serum concentrations of glial fibrillary acidic protein and neurofilament light chain was measured in each patient. The authors conclude that the serum glial factor concentration increased, and the degree of increase was that corelated with infarct size. The study seems carefully performed and the conclusions are appropriate. Cardiac MRI was used for infarct size evaluation and is clearly a strength of the study. The manuscript is well written and easy to read. Additional measurements in serum samples could increase the scientific significance.

Most myocardial infarcts are caused by atherosclerosis, a condition that develops with age as a systemic disease and according to this study, in parallel with increasing concentrations of GFAT and NfL. Values were given in pg/ml. To propose that an increase in production of GFAT is the result of MI it require that all the factors that determine intravascular volume remain constant after myocardial infarction and that GFAT metabolization is constant. Alternative standardization (comparable to tissue immune assays) relative to other protein components in serum should be tested. Also, it would strengthen the study if proteins related to systemic inflammatory status during the post-infarct follow up was reported.

Author Response

The current manuscript describes one year follow up of patients with first ST-elevation myocardial infarction. The degree of change in serum concentrations of glial fibrillary acidic protein and neurofilament light chain was measured in each patient. The authors conclude that the serum glial factor concentration increased, and the degree of increase was that correlated with infarct size. The study seems carefully performed and the conclusions are appropriate. Cardiac MRI was used for infarct size evaluation and is clearly a strength of the study. The manuscript is well written and easy to read.

Thank you for this constructive feedback. Indeed, cardiac MRI enabled a detailed characterization of parameters like ejection fraction and infarction size after myocardial infarction, which is a strength of this study.

Additional measurements in serum samples could increase the scientific significance. Most myocardial infarcts are caused by atherosclerosis, a condition that develops with age as a systemic disease and according to this study, in parallel with increasing concentrations of GFAT and NfL. Values were given in pg/ml. To propose that an increase in production of GFAT is the result of MI it require that all the factors that determine intravascular volume remain constant after myocardial infarction and that GFAT metabolization is constant. Alternative standardization (comparable to tissue immune assays) relative to other protein components in serum should be tested. Also, it would strengthen the study if proteins related to systemic inflammatory status during the post-infarct follow up was reported.

Thank you for addressing this important issue. To clarify the point you raised, we now checked, how other laboratory parameters changed in the year after MI. We included this information as Table 3 in our manuscript: Indeed, we detected significant increases of natrium, potassium, uric acid, high density lipoprotein, triglycerides, albumin, hemoglobin and hematocrit in the 12 months after MI. In contrast, estimated glomerular filtration rate, cholesterol, low density lipoprotein, lactate dehydrogenase, creatine kinase, NT-proBNP, troponin T, thrombocytes, leucocytes and C-reactive protein declined in this period of time:

Table 3. Course of laboratory parameters after first STEMI and correlation to ΔGFAP. * = significant change in Wilcoxon matched-pairs signed-ranks test

Day 0-4

12 months

Δ12m – d0-4

~ ΔGFAP

p value

Glial fibrillary acidic protein (pg/ml)

64 (47, 90)

73 (53, 113)

10 (1, 23)*

-

Neurofilament light chain (pg/ml)

11 (7, 15)

10 (7, 14)

0 (-2, 1)

0,32*

0.030

Natrium (mmol/l)

139 (137, 140)

140 (138, 141)

1 (-1, 2)*

-0,14

ns

Potassium (mmol/l)

4.2 (4.0, 4.4)

4.4 (4.2, 4.6)

0.2 (0.0, 0.4)*

0,03

ns

Estimated GFR (ml/min/1.73 m²)

91 (80, 106)

82 (72, 99)

-7 (-14, 3)*

-0,14

ns

Urea (mg/dl)

31 (25, 38)

31 (26, 37)

0 (-7, 7)

0,13

ns

Uric acid (mg/dl)

5.2 (4.5, 6.0)

6.0 (5.0, 6.6)

0.7 (0.0, 1.4)*

0,17

ns

Cholesterol (mg/dl)

190 (165, 211)

161 (145, 190)

-25 (-60, 11)*

-0,08

ns

Low density lipoprotein (mg/dl)

95 (78, 114)

82 (66, 101)

-11 (-42, 5)*

0,03

ns

High density lipoprotein (mg/dl)

39 (33, 46)

45 (38, 59)

5 (2, 9)*

-0,02

ns

Triglycerides (mg/dl)

128 (105, 162)

162 (98, 195)

45 (-12, 108)*

-0,14

ns

Aspartate aminotransferase (U/l)

48 (33, 93)

28 (21, 32)

-21 (-155, -4)

-0,07

ns

Alanine aminotransferase (U/l)

45 (31, 65)

30 (20, 34)

-19 (-27, 3)

-0,23

ns

γ-glutamyltransferase (U/l)

44 (28, 70)

32 (19, 38)

-3 (-53, 4)

-0,22

ns

Lactate dehydrogenase (U/l)

530 (366, 657)

198 (175, 240)

-293 (-491, -108)*

-0,15

ns

Creatine kinase (U/l)

560 (302, 1136)

113 (82, 165)

-426 (-975, -95)*

-0,19

ns

NT-proBNP (pg/ml)

844 (627, 1368)

133 (48, 291)

-755 (-1329, -403)*

-0,02

ns

Troponin T (pg/ml)

2044 (950, 2952)

6 (5, 9)

-2037 (-2946, -945)*

-0,03

ns

Complement factor C3c (mg/dl)

121 (107, 132)

118 (105, 131)

-3 (-14, 8)

-0,02

ns

Albumin (g/dl)

4.2 (3.9, 4.3)

4.6 (4.4, 4.9)

0.5 (0.3, 0.8)*

0,00

ns

Hemoglobin (g/dl)

13.9 (13.0, 15.1)

14.6 (13.8, 15.5)

0.8 (0.2, 1.7)*

-0,09

ns

Hematocrit (%)

41 (38, 43)

43 (41, 45)

3 (0, 5)*

-0,16

ns

Thrombocytes (10³/µl)

242 (189, 288)

235 (189, 285)

3 (-23, 19)

0,12

ns

Leucocytes (10³/µl)

9.2 (8.1, 10.9)

6.3 (5.2, 7.2)

-2.8 (-4.2, -1.9)*

-0,14

ns

HbA1c (%)

5.6 (5.4, 6.7)

5.7 (5.4, 6.0)

0.0 (-0.2, 0.3)

0,15

ns

C-reactive protein (mg/dl)

1.7 (0.7, 6.2)

0.1 (0.0, 0.2)

-1.6 (-6.1, -0.6)*

0,12

ns

In a next step, we tested, whether these laboratory alterations relate to changes of GFAP. Therefore, we performed Spearman’s correlation (last column): None of the investigated parameters (including albumin) significantly correlated to changes of GFAP. Only ΔNfL positively related to ΔGFAP, underscoring some parallel behavior of both biomarkers. We added this information to the results section on page 4:

To exclude that changes of other laboratory parameters are involved in the reported increase of GFAP, we analyzed how available values changed in the year after MI (Table 3): Upon other, we detected increases of hemoglobin and albumin and decreases of C-reactive protein and glomerular filtration rate. Importantly, these changes did not correlate to changes of GFAP (ΔGFAP) one year after MI.

Moreover, as you mentioned, the extend of systemic inflammation after MI might trigger increases of GFAP. This is why we tested, whether peak C-reactive protein and peak leukocytes relate to absolute and differential values of GFAP and NfL. As now depicted in Table 2, this was not the case in the current analysis.

Reviewer 2 Report

In order to become a more logically strong paper, additional explanations are needed compared to the biomarkers identified in the current study and other biomarkers identified in the past.

Author Response

In order to become a more logically strong paper, additional explanations are needed compared to the biomarkers identified in the current study and other biomarkers identified in the past.

This is a very important point, thanks for bringing this up.

To our knowledge, this is the first report on the course of the neurological biomarkers NfL and GFAP after myocardial infarction. Reports on neuron-specific enolase (NSE) after MI without resuscitation are also missing. When looking for reports on other potential neurological biomarkers after MI, we found the following investigaton, which we included in our discussion on page 6:

Interestingly, a recent study found that plasma S100 calcium-binding protein B (S100B) levels increased significantly in acute MI patients compared to the levels of stable angina pectoris patients and control subjects [34]. As S100B is also considered a peripheral marker of blood-brain barrier disruption [35], these findings may suggest that a disruption of this structure after MI causes the reported increase of GFAP.

While Ubiquitin Carboxyl-Terminal Hydrolase L1 (UCHL1) is also considered a peripheral marker of neuronal damage, an upregulation of UCHL1 in post-MI hearts occurs primarily in the cardiomyocytes and protects against post-MI cardiac remodeling (PMID: 35463782). Therefore, it seems inapplicable as a potential specific neuronal marker after MI.

Reviewer 3 Report

I have read the manuscript with interest. The topic is interesting, study conduct and prsentation of results are well done. I don't have any specific comments to the authors and I don't think that any revision is neccessary. To summarize : good job, well done.

Author Response

I have read the manuscript with interest. The topic is interesting, study conduct and presentation of results are well done. I don't have any specific comments to the authors and I don't think that any revision is necessary. To summarize: good job, well done.

                Thank you for this very positive response.
